# Comparative transcriptome analysis of albino northern snakehead (*Channa argus*) reveals its various collagen-related DEGs in caudal fin cells

Shixi Chen[1,2☯], Ning Li[3☯], Fardous Mohammad Safiul Azam[1,4]*, Li Ao[1], Na Li[1], Jianlan Wang[1], Yuanchao Zou[1,2]*, Rui Li[1], Zakaria Hossain Prodhan[1]

1 College of Life Sciences, Neijiang Normal University, Neijiang, China, 2 Conservation and Utilization of Fishes resources in the Upper Reaches of the Yangtze River, Key Laboratory of Sichuan Province, Neijiang, China, 3 Sichuan Yukun Aquatic Technology Co., Tongchuan District, Dazhou City, Sichuan Province, China, 4 Department of Biotechnology and Genetic Engineering, Faculty of Life Sciences, University of Development Alternative, Dhaka, Bangladesh

☯ These authors contributed equally to this work.
* shojibbiotech@yahoo.com (FMSA); zou3891@163.com (YZ)

**Data Availability Statement:** The data supporting these findings is available in online repositories. The names of the repository/repositories and

## Abstract

The albino northern snakehead (*Channa argus*) is an aquaculture species characterized by heritable albino body color, in contrast to the typical coloration. Additionally, there are gray- and golden-finned individuals, which exhibit distinct coloration in their caudal fins. We performed RNA-seq to profile the transcriptome of caudal fin tissues in albino gray-finned and golden-finned *C. argus*, contrasting these with normal morphs to elucidate the differences between the two groups. A total of 137,130 unigenes were identified in this study. Gene Ontology (GO) analysis showed that the identified DEGs were significantly enriched in cellular components related to cytoplasm. So far, 379 common DEGs have been identified in all three groups. Notably, we observed more DEGs in golden-finned individuals compared to gray-finned individuals. We also revealed that golden-finned individuals were enriched in collagen-related pathways compared with normal individuals. The enriched DEGs of collagen components include collagen I of *COL1A1* and *COL1A2*, collagen II of *COL2A1*, collagen V of *COL5A1* and *COL5A2*, collagen VI of *COL6A1* and *COL6A3*, collagen IX of *COL9A3*, collagen X of *COL10A1*, collagen XI of *COL11A2*, collagen XII of *COL12A1*, collagen XVI of *COL16A1*, collagen XVIII of *COL18A1* and decorin (DCN), all of which play a role in modulating the collagen matrix. In golden-finned albino fish, collagen-related genes were downregulated, suggesting that despite the abundance of collagen types in their caudal fin cells, gene expression was slightly limited. This work provides valuable genetic insights into collagen variation in albino *C. argus*, lays the foundation for research on collagen genes and is crucial for the development and utilization of fish-derived collagen as a biomaterial for tissue engineering and biomedical applications.

accession numbers are as follows: Bioproject number PRJNA913664 and BioSample number SAMN32303246 with accession number of SRX18894708-SRX18894716.

**Funding:** This study was funded by the Natural Science Foundation of Sichuan Province (http://202.61.89.120/) (No. 2022NSFSC1721 to SC, 2021YFN0033 to YZ), Central Guided Local Science and Technology Development Funding (23ZYZYTS0356 to SC) of Science and Technology Department of Sichuan Province (http://202.61.89.120/), Science and Technology Program Projects of Sichuan Provincial Science and Technology Department (2021YFN0028 to RL) ((http://202.61.89.120/)), Neijiang Normal University Science and Technology Foundation (http://www.njtc.edu.cn) (2019FM05 to SC), and the Neijiang Normal University School-level Research project (http://www.njtc.edu.cn) (2023YB12 to LA). The funders had no role in study design, data collection and analysis, decision to publish, or preparation of the manuscript.

**Competing interests:** The authors have declared that no competing interests exist.

# 1 Introduction

Collagens, the most abundant proteins in the animal kingdom, are best recognized for providing mechanical strength to skin, bones, and other tissues [1]. It plays a key role in retaining the texture of fish muscles [2, 3], and increased collagen content in muscle enhances the firmness of fish fillets [4, 5], which is a vital feature in determining the acceptability of aquatic products. Owing to high water absorption capacity, high biocompatibility, low immunogenicity, biodegradability, high porosity, ease of processing, ability to penetrate a lipid-free interface, limited disease transmission, natural ability to combine with other materials (synthetic polymers), and options for industrial-scale extraction, collagen has limited or no ethical or religious constraints. Because of these properties, collagen has become a key multifunctional protein used in the food, cosmetics, pharmaceutical and biomedical industries [6]. Compared with porcine and bovine collagen is more readily absorbed and has higher bioavailability [7]. Altogether, they have become valuable biomaterials and have received significant attention from biomedical researchers [8]. In its therapeutic aspects, collagen derivatives are used for tissue engineering, cell culture, and drug delivery and have shown an ability to inhibit bacterial growth and promote wound healing [9, 10]. For instance, wound contraction rate and histological examination demonstrated that the prepared collagen effectively promoted wound healing, resulting in improved wound resolution and closure [9, 11]. The evidence reviewed above suggests that fish collagen is a promising option for wound healing and other therapeutic applications. In general, three collagen chains are assembled into a left-handed triple helix to form a protein complex, which is then built into higher-order structures [12, 13]. To date, 28 distinct forms of collagen (collagen I–XXVIII) have been identified [13–15]. In fish, type I collagen is the most frequent type; however, certain species contain type II collagen [10].

The northern snakehead (*Channa argus*) is a Channa species from the Channidae family of Perciformes, found in Asia and Africa [16, 17]. Moreover, in China, the annual production of this species is estimated to be approximately 510,000 tons, with a value of 1.6 billion US dollars [18]. However, it is classified as an invasive species in North America that threatens many other species [19]. This species is the most important snakehead species, with only a few bone spurs cultured in China and Malaysia. It has an edible rate of up to 63% and has gained popularity among people in these countries due to its high nutritional content [20–22]. It is also sold in ethnic live-food fish markets in New York and St. Louis, Missouri [17]. The northern snakehead albino phenotype is a unique heritable albino that is found exclusively in China. It was first recorded in Neijiang City, Sichuan Province, China [23, 24]. The natives of this city believe that albino *C. argus* with golden caudal fins has medicinal value, such as healing wounds and promoting lactation [25]. Considering its unique features and significant uses, several molecular studies have been conducted, including studies on the draft whole genome [18], genome characterization [26], molecular diversity [27], disease and pathogenesis [28], skin color and albinism [29], sex chromosomes [30], feeding behavior [24, 31] and surimi products [32].

Transcriptome sequencing analysis is widely utilized to identify differentially expressed genes (DEGs) and further investigate the molecular mechanisms involved in metabolic pathways [33–35]. Multiple studies have identified DEGs associated with collagen metabolism, such as palatal organ development in bighead carp (*Hypophthalmichthys nobilis*) [36], flesh pigmentation in Atlantic salmon (*Salmo salar*) [37], collagen synthesis genes for connective tissue growth factor (*Nibea coibor*) [38], and collagen deposition in zebrafish [3]. DEG analysis, for example, has shown the up- and downregulation of skin color genes, as well as the identification of five causative genes for albinism in this species via whole transcriptome sequencing [39]. To further elucidate the mechanism of albino features in albino-type (AT)

and bicolor-type (BT) fish, histological and transcriptome analyses were conducted to identify pigment-related genes in *C. argus* [40]. Their study uncovered a comprehensive process in the regulation of genes associated with color and albinism in the skin and spine tissues of *C. argus*. Nevertheless, the distinctive golden-colored caudal fin trait has hitherto not been explored in transcriptome studies. In addition, the hunt for novel collagen sources from fish, particularly fins, is a growing trend in the development of biomaterials for medical and tissue engineering applications [41, 42]. Thus, studying collagen metabolism in *C. argus* caudal fins might lead to the discovery of a novel biomaterial source since albino *C. argus* with golden caudal fins possesses wound healing capabilities that set it apart from the gray-finned albino. To the best of our knowledge, no research has been performed on collagen metabolism in northern snakeheads. This is the first report on albino *C. argus*, which revealed the presence of collagen-rich caudal fin cells.

## 2 Materials and methods

### 2.1 Experimental fish

The fish used in this study were healthy, one-year-old *C. argus* collected from the Guiming Aquaculture Family Farm in Chaoyang Town, Neijiang City, Sichuan Province, China. Caudal fin tissues were obtained from nine *C. argus* (three repeats per group), comprising the W group of three black-white *C. argus*, H group of three gray-finned albino *C. argus*, and J group of three golden-finned albino *C. argus* (Fig 1). The tissues from the three replicates were collected in liquid nitrogen to generate three biological replicates, which were then preserved in a -80°C freezer until needed. The study was approved by the Ethics Committee of Neijiang Normal University (protocol code SK1729 and approved on 2021-10-05). All methods were performed following relevant guidelines and regulations. The study was conducted under the ARRIVE guidelines [43].

### 2.2 RNA extraction, library construction and sequencing

Caudal fin tissue from each sample was crushed into a powder in liquid nitrogen, and total RNA was extracted using TRIzol reagent (Invitrogen, USA) according to the manufacturer's instructions. RNA quality and concentration were measured via 1% agarose gel electrophoresis and a Nanodrop (Thermo Fisher Scientific, Waltham, MA). A Fragment Analyzer 5400

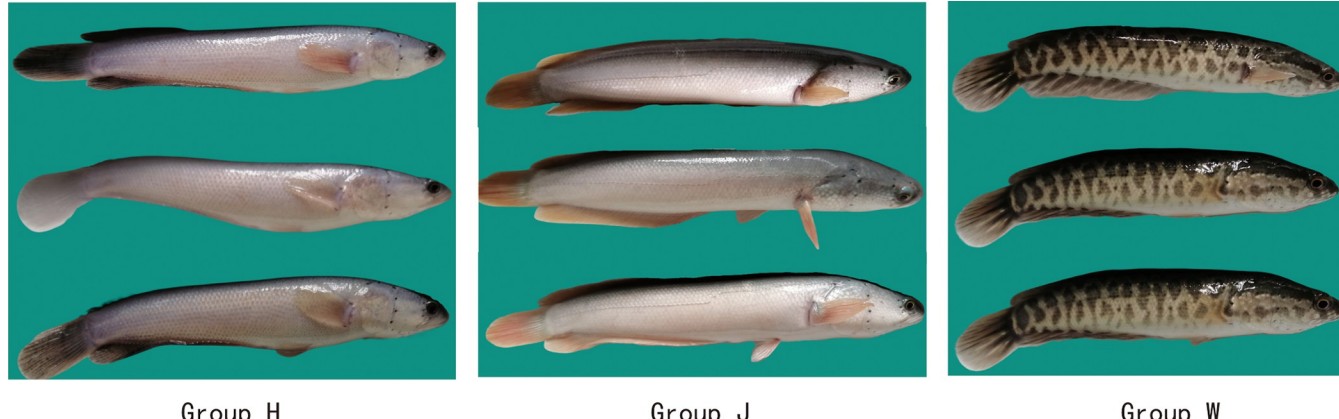

Group H          Group J          Group W

**Fig 1. *Chana argus* was used in this study.** Three distinct color morphs of *C. argus* were identified: Group H included three gray-finned albinos, Group J included three golden-finned albinos, and Group W included three black-whites.

(Agilent Technologies, CA, USA) was used to analyze RNA integrity. Index-coded samples were clustered using a cBot Cluster Generation System with a TruSeq PE Cluster Kit v3-cBot-HS (Illumina Inc., San Diego, CA, USA) according to the manufacturer's instructions. Next, we sequenced the library preparations using an Illumina NovaSeq 6000 platform and generated 150 bp paired-end reads.

## 2.3 Bioinformatics analysis of the nonparticipating transcriptome

The original fluorescence image files obtained from the Illumina platform were transformed into short reads (raw data) by base calling, and these short reads were recorded in FASTQ format, which includes both sequence and sequencing quality data [44]. Raw reads were trimmed to obtain high-quality, clean reads. Trimmomatic software was used to filter the sequencing data [45] and clean reads were assembled via Trinity [46]. The assembly quality of Trinity. fasta, unigene.fa and cluster.fasta was evaluated using the BUSCO software (S1 Fig) [47]. The unigene sequence acquired by Trinity assembly was used as the reference sequence, and the clean reads from each sample were compared to Ref., filtering out the reads with a comparison quality of less than 10, non-comparable reads, and reads compared to multiple regions of the genome. We extracted and merged the read count data to get the read count values for each sample, and the transcripts per kilobase million (TPM) values were computed using RNA-Seq via expectation maximization (RSEM) software with the default threshold. We collected and combined TPM data from each sample to determine gene expression levels [48]. Davidson and Oshlack [49] used the Corset to perform hierarchical transcript clustering. DEGs were defined as having an adjusted P-value <0.05, and a |log2(fold change)| > 1. To detect DEGs in the samples, we performed DESeq2 at different time points compared with the control group [50]. We utilized Cluster Profiler to analyze DEG enrichment in Gene Ontology (GO) and the Kyoto Encyclopedia of Genes and Genomes (KEGG) [51, 52].

## 2.4 Phylogenetic tree construction

Each collagen-related transcript identified in *C. argus* was subsequently analyzed using BLASTn to search for homologous genes in the NCBI database (https://blast.ncbi.nlm.nih.gov/Blast.cgi?PROGRAM=blastn&PAGE_TYPE=BlastSearch&LINK_LOC=blasthome; accessed date: April 5, 2024). For the phylogenetic analysis, we selected the top 15 homologous genes from the search results for each collagen gene (S1 Table). Multiple sequence alignments were carried out, and phylogenetic analysis was performed using MEGA11 software to construct a neighbor-joining phylogenetic tree, which was validated by a 1000-iteration bootstrap test [53]. Later, the linked fish species and families of each homologous gene were examined to ascertain the prevalence of each collagen-related gene within these families. The data were then tallied and shown as a stack bar graph.

## 3 Results

### 3.1 Transcriptome assembly

The current study categorized the *C. argus* specimens into three groups: H, J, and W. Groups H and J possessed hereditable white body coloring of *C. argus*, whereas group W had both black and white coloration. The disparity in body coloration among the three groups was significant at the caudal fin, with group J differing from the other two by its golden hue. Despite the sequencing of three transcriptome sets, the ensuing research primarily concentrated on the variation between groups W and J due to their pronounced body color features and transcriptome variations.

**Table 1. Sequencing data list of *Channa argus*.**

| mple id | raw base (Gb) | raw sequences | clean reads (Gb) | clean bases | q30 (%) | GC (%) | Assembled sequences | unigene |
|---|---|---|---|---|---|---|---|---|
| H1 | 6.26 | 41704178 | 5.83 | 39432270 | 99.9 | 45.5(%) | | |
| H2 | 6.39 | 42584530 | 5.97 | 40468492 | 99.9 | 46.0(%) | | |
| H3 | 7.46 | 49740560 | 6.97 | 47218788 | 99.9 | 46.0(%) | | |
| J1 | 6.27 | 41829506 | 5.87 | 39783606 | 99.91 | 46.0(%) | | |
| J2 | 6.53 | 43564594 | 6.1 | 41357768 | 99.9 | 46.0(%) | | |
| J3 | 6.37 | 42467434 | 5.93 | 40201484 | 99.9 | 46.0(%) | | |
| W1 | 5.97 | 39784420 | 5.56 | 37660476 | 99.9 | 45.0(%) | | |
| W2 | 6.2 | 41312750 | 5.8 | 39229648 | 99.9 | 45.5(%) | | |
| W3 | 6.48 | 43183456 | 6.09 | 41173154 | 99.91 | 45.0(%) | | |
| **Total** | **57.93** | | **54.12** | | | | **174779** | **137130** |

For each fish examined, transcriptome sequencing of *C. argus* produced a raw base of 5.97 to 7.46 Gb; the quantity of clean data was 54.12 Gb, including 174,779 assembled sequences, 137,130 unigenes, 86827 predicted coding sequences (CDSs) and 2,532 clusterSeqs. The GC content of each transcriptome ranged from 45 to 46%. A summary of the caudal fin transcriptome data for *C. argus* is presented in Table 1.

Five functional databases were used to classify these unigenes, and 86,828 NR (63.32%), 105,747 NT (77.11%), 71,814 SwissProt (52.37%), 70,693 KOG (51.55%), and 71,751 UniProt (52.32%) genes were identified; 68,700 of these unigenes were commonly annotated, and 56,615 KEGG (41.29%) and 55,175 GO (40.24%) were annotated (Fig 2A). The stronger the correlation coefficient between the samples, the more similar their expression patterns. The samples for the H, J and W groups were 0.9 ~ 1.0 and were closely correlated (Fig 2B). The sequencing data generated in this study were deposited in the Sequence Read Archive of the National Center for Biotechnology Information (NCBI) (accession number: PRJNA913664).

## 3.2 Functional annotation of DEG genes and their variations

Gene Ontology (GO) annotation and enrichment analysis of the DEGs in *C. argus* demonstrated that the DEGs were associated with the terms biological process (BP), cellular component (CC), and molecular function (MF), with the most enriched term being the processes related to the cytoplasm of CC (Fig 3).

We identified and clustered them to investigate variations in DEG expression modalities. Expression patterns of the same gene clusters were comparable. DEGs were significantly separated into groups J, H and W as shown in the heatmap. Group J was in a separate cluster compared to groups H and W, indicating a greater disparity between the two groups (Fig 4).

The overlap of DEGs across different comparison combinations is depicted in a Venn diagram (Fig 5A). This study compared the DEGs of groups J or H with those of group W to identify common DEGs, revealing 2098 DEGs in group J, 1603 DEGs in group H, and a total of 379 common DEGs. The GO enrichment analysis indicated that the quantities of upregulated genes, downregulated genes, and total genes within each of the three GO categories (cellular component, biological process, and molecular function) were counted individually and are displayed as bar graphs (Fig 5B and 5C). The results revealed that both groups H and J exhibited more downregulated DEGs compared to group W.

The heatmap and Venn diagram revealed that the DEGs in group J compared to group W exceeded those in group J compared to group H. Consequently, the following analysis was conducted between groups J and W.

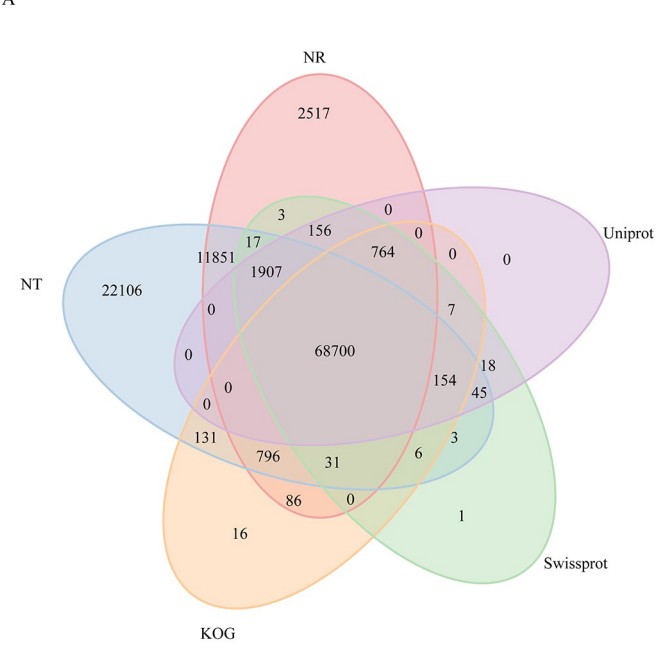

**Fig 2.** A. Venn diagram based on 5 databases: NR, NT, SwissProt, KOG and UniProt; numbers in the overlapping regions refer to those unigenes that were expressed in more than one database, and 68700 shared unigenes matched with all five databases; B. Correlations of the samples between the H (H1, H2, H3), J (J1, J2, J3) and W (W1, W2, W3) groups. The change in the correlation coefficient is indicated by the change in color from blue (1.0) to red (-1.0), and a deeper color indicates a higher correlation between samples.

### 3.3 Distinct collagen-related pathways in white *C. argus* with golden fins

Gene Ontology (GO) enrichment analysis confirmed that DEGs from group J, compared with W, were significantly enriched in collagen-related pathways (p-value < 0.05). These pathways

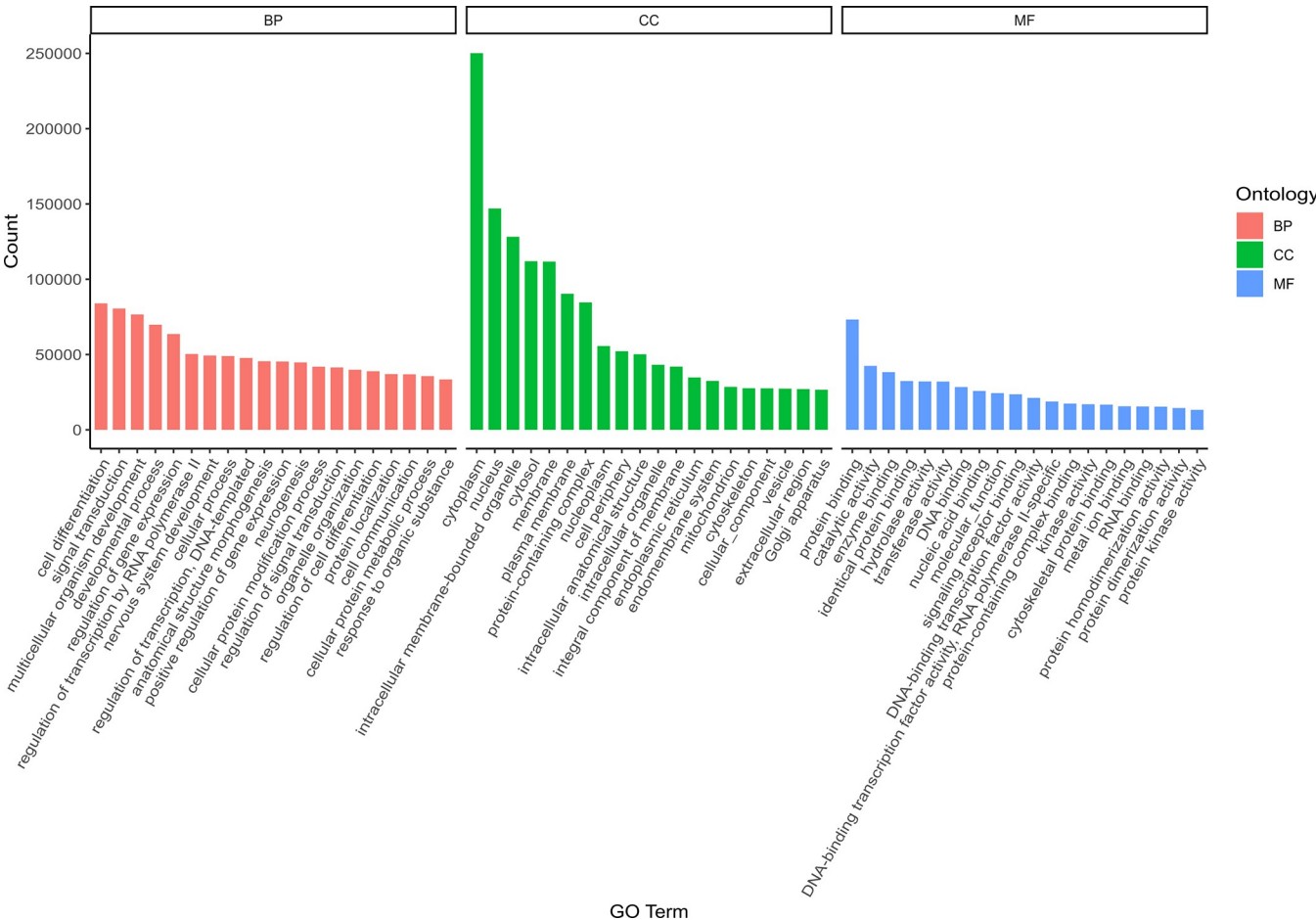

**Fig 3. Genes were classified according to three major GO categories: Biological process (BP), cellular component (CC), and molecular function (MF).**
The y-axis indicates the number of genes in each category.

encompassed collagen trimer, fibrillar collagen, banded collagen fibril, a complex of collagen trimers, collagen-activated tyrosine kinase receptor signaling pathway, collagen-activated signaling pathway and collagen type I trimer (Fig 6). Additional pathways included cellular responses to vitamins, synthesis of molecular mediators involved in the inflammatory response, protein heterodimerization, binding platelet-derived growth factor, tooth mineralization, positive regulation of long-term synaptic depression, cellular responses to vitamin E, CDP-alcohol phosphatidyl transferase activity, CDP-diacylglycerol-serine O-phosphatidyl transferase activity, tooth eruption, cardiac neuron differentiation, cardiac neuron development and negative regulation of monocyte chemotactic protein-1 production.

The directed acyclic graph illustrates the outcomes of the differential gene GO enrichment analysis, with branches representing inclusion relationships and progressively specific functional ranges defined from top to bottom (Fig 7). Colored shades denote the degree of enrichment, with darker colors indicating higher enrichment. The J-W CC top GO-directed acyclic network showed distinctive enrichment of the cellular component (1145/52530). The collagen components included a collagen-containing extracellular matrix (60/711), a complex of collagen trimers (16/69), collagen trimers (28/169), fibrillar collagen trimers (16/39), banded collagen fibrils (16/39), collagen type I trimers (7/10), and collagen type V trimers (5/17).

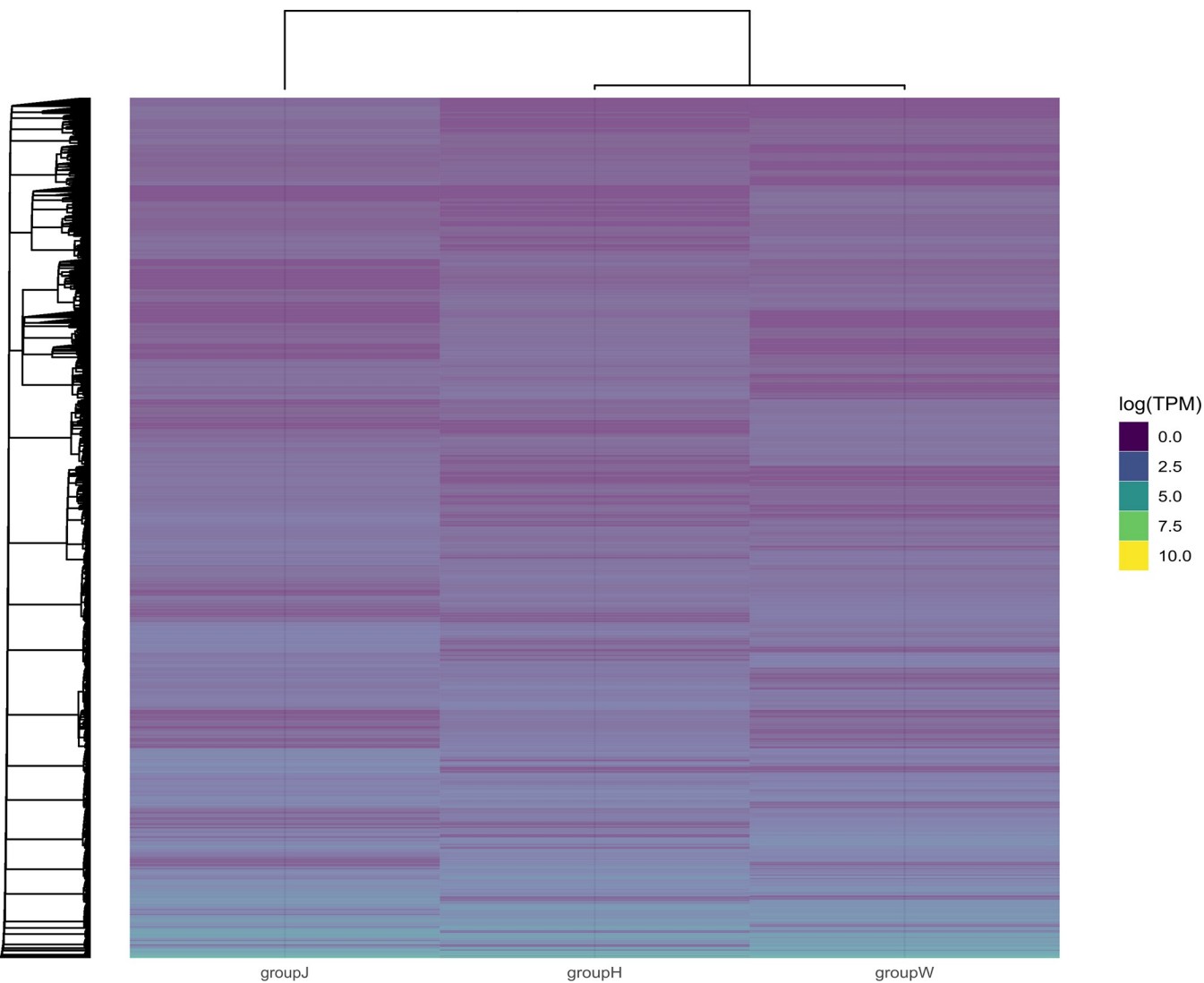

**Fig 4. Heatmap of differentially expressed gene clustering.** The colors can only be compared horizontally with the same clustering of genes for expression values (TPM) in the high (blue) and low (yellow) colors. Axes: Bottom, variable groups; top, hierarchical clusters for closeness between groups; left, hierarchical clustering among the expressed genes.

The DEGs associated with collagen component terms included collagen I of *COL1A1* and *COL1A2*, collagen II of *COL2A1*, collagen V of *COL5A1* and *COL5A2*, collagen VI of *COL6A1* and *COL6A3*, collagen IX of *COL9A3*, collagen X of *COL10A1*, collagen XI of *COL11A2*, collagen XII of *COL12A1*, collagen XVI of *COL16A1*, collagen XVIII of *COL18A1* and decorin (DCN) for modulation of the collagen matrix. The DEGs of the J group exhibited a greater abundance of collagen component terms than those in the W group.

The analysis of log2-fold changes in expression levels among the collagen-related genes included six transcripts for *COL1A1*, one for *COL1A2*, two for *COL2A1*, one for *COL5A1*, four for *COL5A2*, one for *COL6A1*, four for *COL6A3*, one for *COL9A3*, one for *COL10A1*, one for *COL11A12*, two for *COL12A1*, one for *COL16A1* and one for *COL18A1* (Fig 8). In comparing the log2-fold changes in expression levels between the W and J groups, the J group was found to have significantly downregulated for collagen-related genes. *COL2A1*, *COL10A1*, *COL11A2*, and *COL18A1* exhibited a downward trend exceeding threefold in the J group.

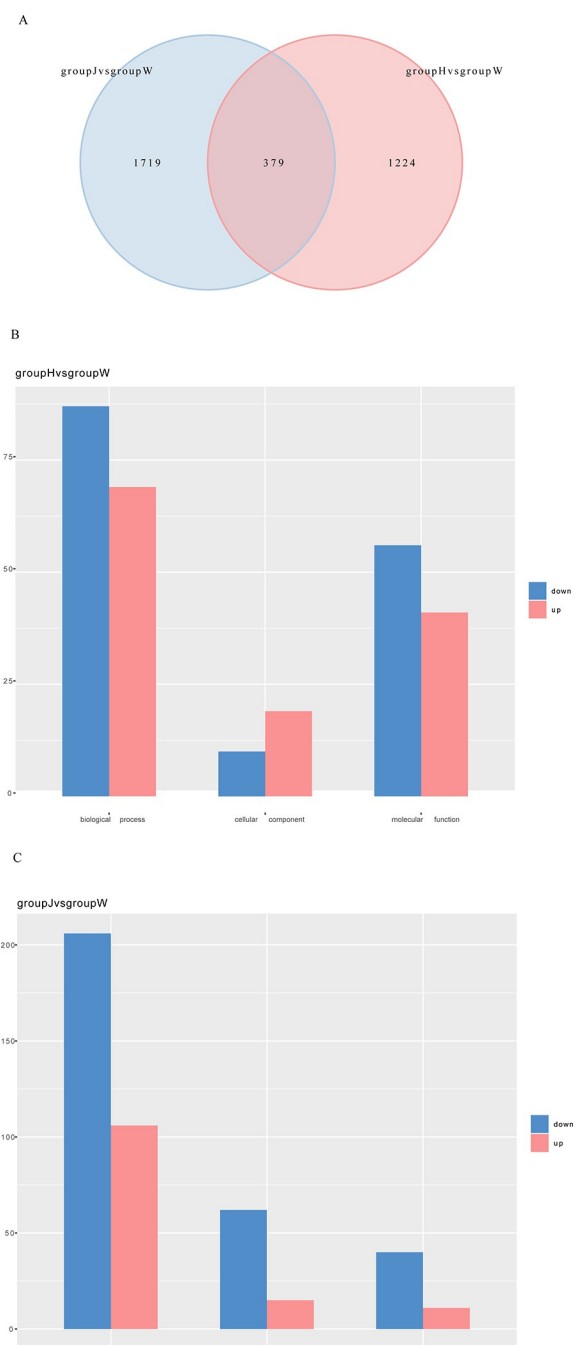

**Fig 5. Differential genetic Venn diagram of groups J and H compared with group W and GO enrichment histogram of differentially expressed genes.** A. Venn diagram between group J vs. group W and group H vs. group W. The number of genes expressed uniquely within each group is shown and overlapping regions indicate the number of genes expressed in both groups. B. Comparison of GO enrichment analysis for up-and down-regulated genes between groups H and W. C. Comparison of GO enrichment analysis for up- and downregulated genes between groups J and W. X-axis: number of genes; Y-axis: GO categories.

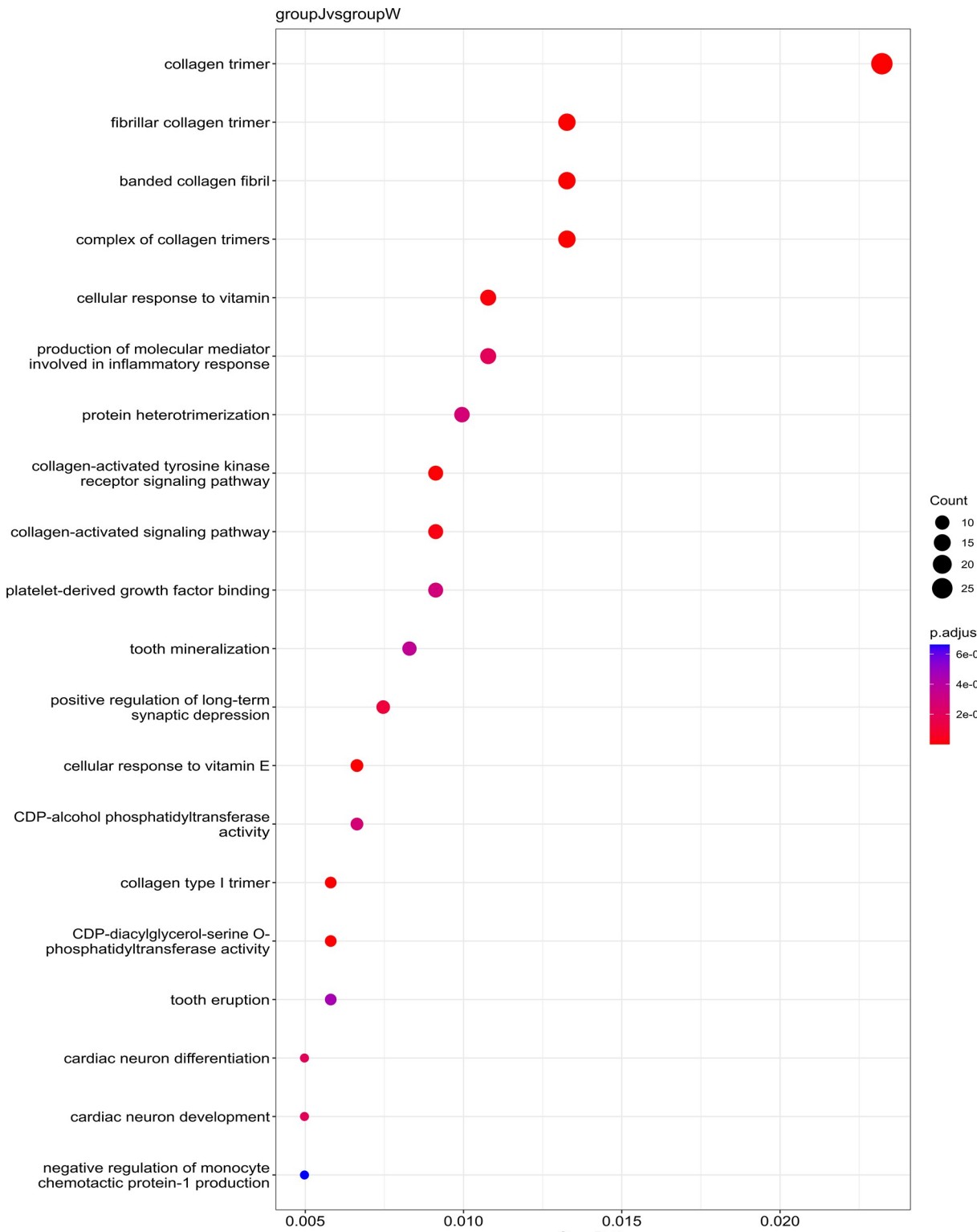

**Fig 6. Bubble diagram of the GO enrichment analysis results for group J compared with those for group W.** The 20 most significantly enriched GO pathway entries are presented in this figure. The value of p-adjust ranges from 0 to 1, and the closer it is to zero, the more significant the enrichment. GeneRatio is the ratio of the number of differentially expressed genes in pathway entry to the total number of genes involved in pathway entry for all annotated genes.

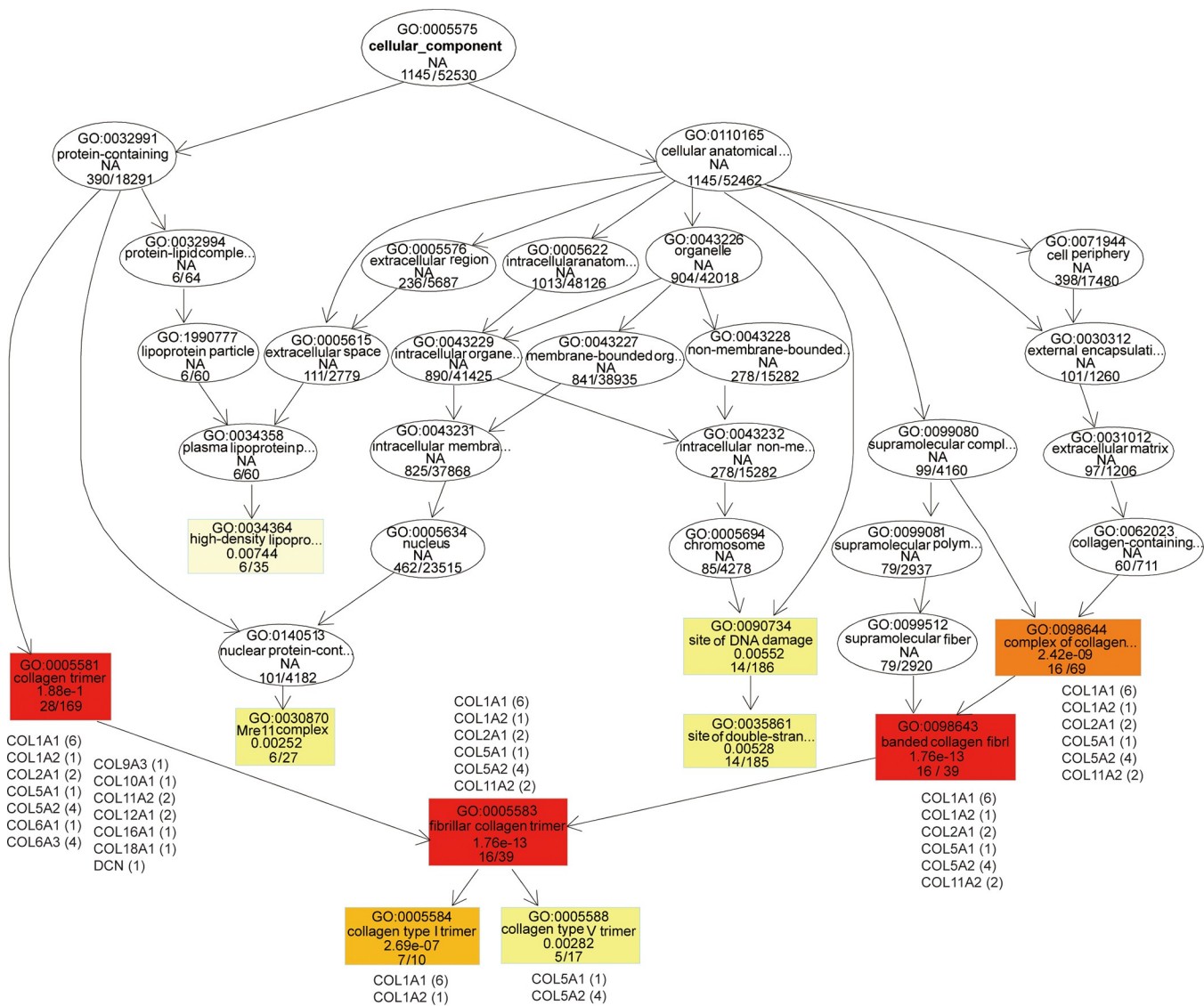

**Fig 7. Directed acyclic graph (DAG) of topGO in group J compared to group W for classification of CCs (cell components).** The branches represent inclusion relationships and the functional descriptions defined from top to bottom become increasingly specific. Each node represents a GO term. The specific DEGs involved in enrichment are shown alongside functional descriptions. The most important terms are represented by red nodes, whereas the less significant terms are represented by elliptical nodes. Colors represent the degree of enrichment, with darker colors representing higher enrichment.

## 3.4 Homologous genes reveal diverse collagen-rich fish families

A homology search in the NCBI database using BLASTn was conducted for each of the 13 collagen-related genes to further predict and understand the relationships among various fish families. A phylogenetic tree was subsequently constructed, revealing two distinct species clusters for each gene (Fig 9). Analysis of the 13 trees indicated that *C. argus* grouped with homologous genes from other species, implying that these sets are notable sources of collagen (Fig 9A–9M). Collagen-specific genes, including *COL1A1*, *COL1A2*, *COL6A1*, *COL6A3*, *COL9A3*, *COL11A1*, *COL16A1* and *COL18A1*, showed a close association between *C. argus* and *Lates calcarifer* (Fig 9A), *Micropterus salmoides* (Fig 9B), *Scatophagus argus* (Fig 9F), *Chelmon rostratus* (Fig 9G), *Seriola lalandi* dorsalis (Fig 9H), *Dicentrarchus labrax* (Fig 9J), *Lates calcarifer*

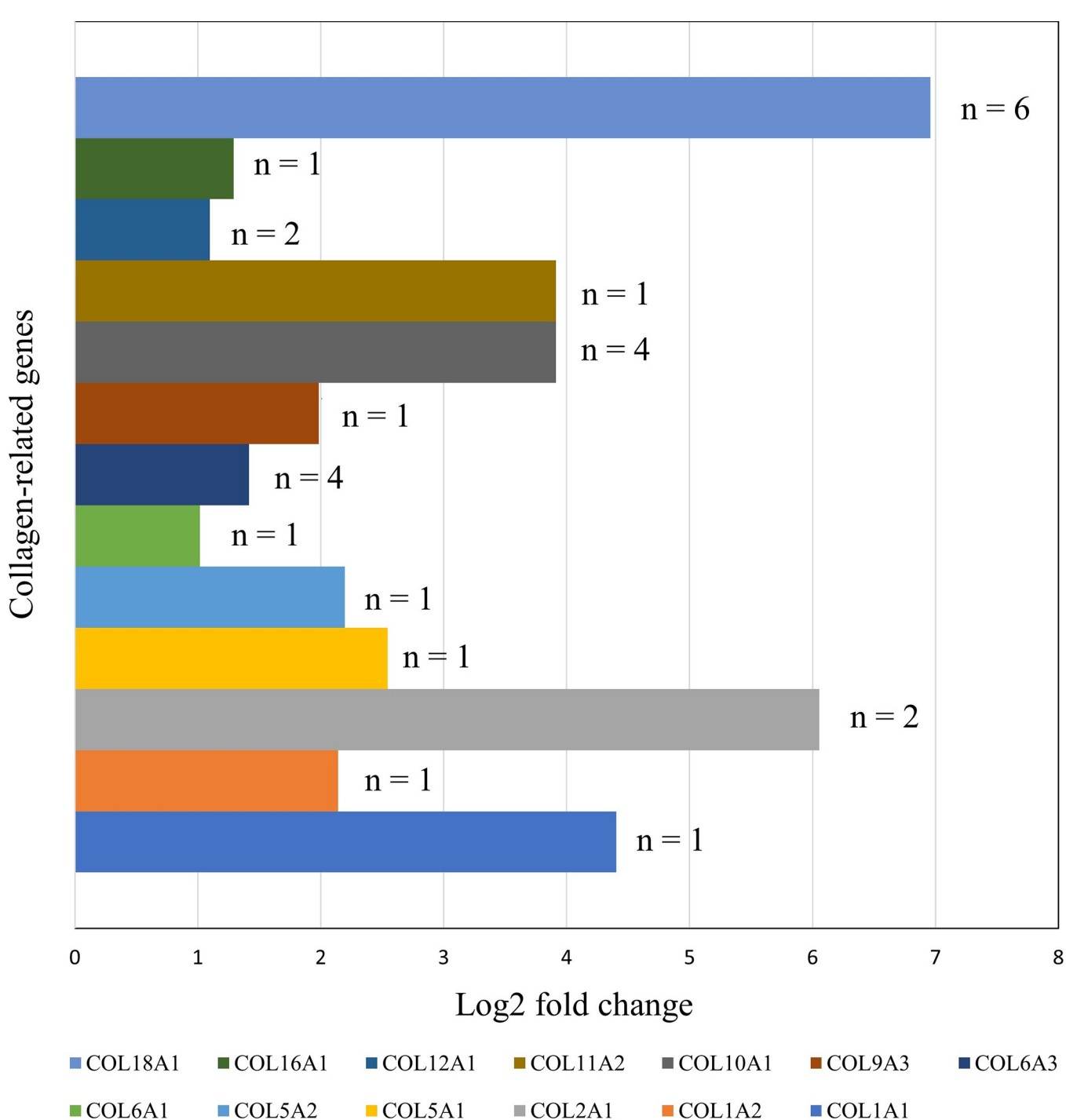

**Fig 8. Downregulation of collagen genes in golden-finned *C argus*.** Axis: x-axis, gene expression level based on log2-fold change value; y-axis, gene names.

(Fig 9L), and *Solea senegalensis* (Fig 9M). Nonetheless, *C. argus* exhibited a distant genetic link with species concerning the genes *COL2A1* (Fig 9C), *COL5A1* (Fig 9D), *COL5A2* (Fig 9E), *COL10A1* (Fig 9I), and *COL12A1* (Fig 9K).

Interestingly, these homologous genes were found across twenty-four fish families derived from 13 phylogenetic trees, each displaying a unique collagen gene variant (S2 Fig). Fig 9

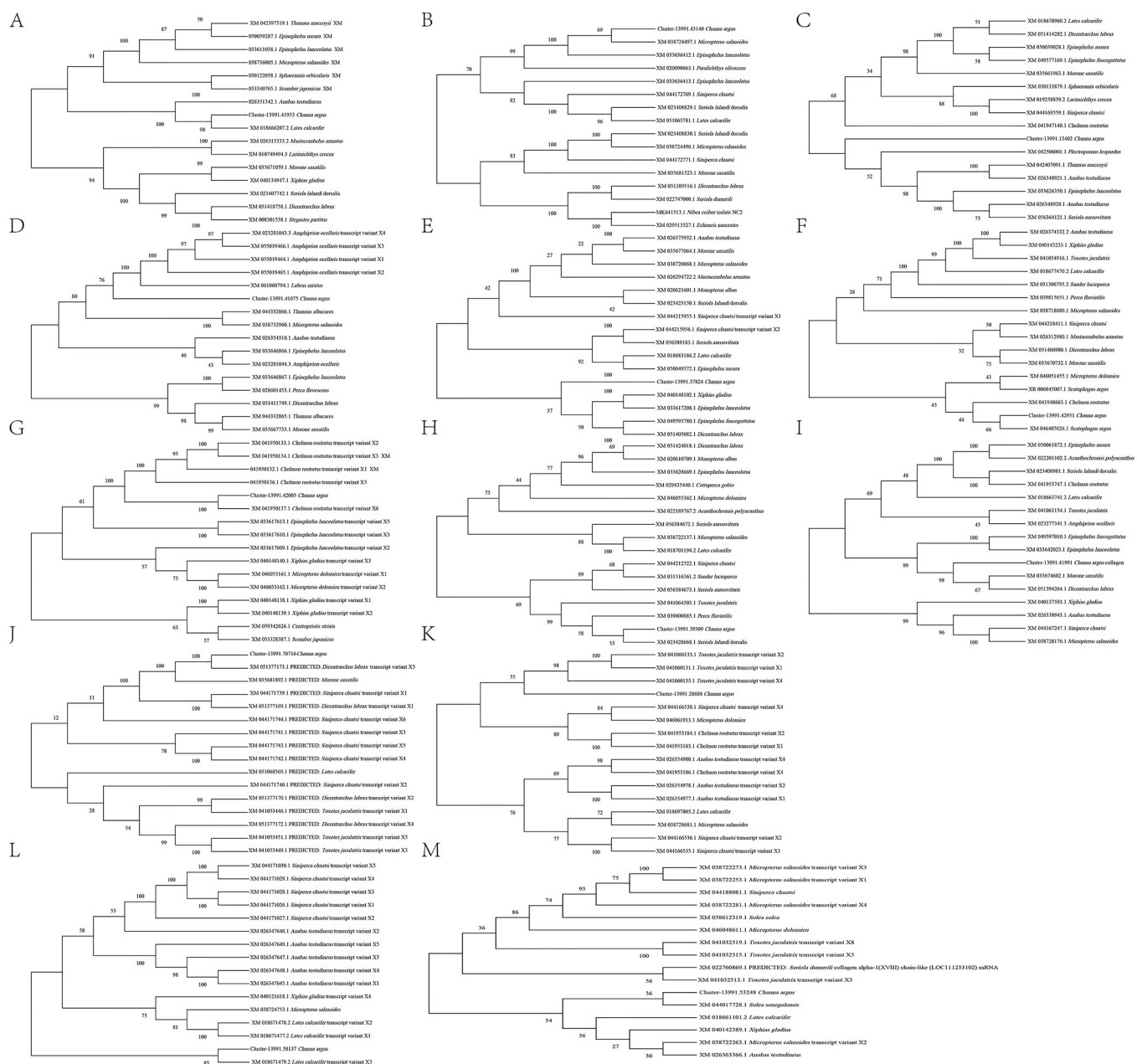

**Fig 9.** (A-M). Phylogenetic studies of collagen-related genes. These correspond to A. COL1A1, B. COL1A2, C. COL2A1, D. COL5A1, E. COL5A2, F. COL6A1, G. COL6A3, H. COL9A3, I. COL10A1, J. COL11A1, K. COL11A2, L. COL16A1 and M. COL18A1.

illustrates the diversity of collagen-related genes in different species among these families including eleven in Centrarchidae and Latidae; ten in Sinipercidae; nine in Anabantidae and Moronidae; eight in Serranidae; seven in Carangidae and Xiphiidae; six in Toxotidae; five in Chaetodontidae; four in Scombridae; three in Mastacembelidae, Percidae, Pomacentridae and Sciaenidae; two in Apogonidae and Synbranchidae; and one in Bovichtidae, Echeneidae, Labridae, Paralichthyidae, Pomacentridae, Scatophagidae and Soleidae. Distinct type of collagen genes were prominent in various fish families, including COL2A1 in Serranidae (n = 4), COL5A1 in Pomacentridae (n = 5), COL6A3 in Chaetodontidae (n = 5) and Serranidae

(n = 4), COL11A2 in Moronidae (n = 5) and Sinipercidae (n = 6), COL16A1 in Anabantidae (n = 5) and Sinipercidae (n = 5), and COL18A1 in Centrarchidae (n = 5).

## 4 Discussion

Numerous species of albino fish exist, including *Silurus glanis*, *Corydoras aeneus*, *Danio rerio*, *Hydrolagus colliei*, *Oryzias latipes*, *Macropodus opercularis*, *Rhamdia quelen*, *Ictalurus punctatus*, *Senegalese sole*, *Scophthalmus maximus*, *Paralichthys olivaceus*, *Triplophysa rosa*, *Carassius auratus*, *Apostichopus japonicus* and *C. argus* [54–63]. Recent genomic research on albino fish, including whole-genome sequencing of snakeheads [64] and RNA sequencing, has been performed to analyze the transcriptome associated with skin pigmentation in this species [39]. Our work presents two phenotypes of heritable albino *C. argus*, designated as groups J and H, and compares their DEGs with those of normal *C. argus* (group W). The body coloration of golden-finned albino *C. argus* in group J was markedly different from that in the normal group. Our study concentrated on three specific color variations in snakehead caudal fins and compared their DEGs, which were similar to those of *Aulonocara baenschi*, an albino cichlid fish [65]. The identified and clustered DEGs provided additional evidence of a significant difference between groups J and W in the present investigation. In comparison with the W group, the DEGs in group J showed mostly enriched cytoplasmic processes inside the cellular component (CC), particularly for collagen components. Likewise, the DEGs in the albino yellow catfish (*Pelteobagrus fulvidraco*) [66] and *Oryzias* species [67] accounted for the highest quantity of genes related to cellular processes and cells and binding cellular components.

Additionally, collagen-related genes were found to be downregulated in the present study, representing a significant discovery in understanding the relationship between albinism and collagen. A recent study reported that albino golden snakehead fish possesses five causative genes associated with albinism, namely *TYR* (tyrosinase), *SOX10* (SRY-box transcription factor 10), *S-100* (S100 calcium-binding protein A1), and *NLRC3* (NLR family CARD domain containing 3), which are significantly downregulated in golden albino *C. argus* [39]. The receptor for advanced glycation end products (RAGE) shares its receptors with the S100 family proteins, collagen (type I and IV) and several other ligands. S100A11 signaling via RAGE activates p38 MAPK kinase, thereby enhancing the synthesis of human collagen X [68]. In the present study on albino snakeheads, the downregulation of *COL10A1* may be linked to S100 proteins, leading to reduced expression of collagen genes, necessitating further investigation for confirmation.

Structural collagen genes are tightly associated with skeletogenesis and are expressed in osteoblasts of fish (gars and zebrafish) [69]. Our study uncovered their complex regulatory mechanism. The DEGs related to the collagen component included *COL1A1* and *COL1A2* for collagen I, *COL2A1* for collagen II, *COL5A1* and *COL5A2* for collagen V, *COL6A1* and *COL6A3* for collagen VI, *COL9A3* for collagen IX, *COL10A1* for collagen X, *COL11A2* for collagen XI, *COL12A1* for collagen XII, *COL16A1* for collagen XVI, and *COL18A1* for collagen XVIII, along with decorin (DCN), which were highly enriched in the caudal fins of golden-finned albino *C. argus*. The set of DEGs identified in this study has not been previously documented in other fish species.

Fish collagen derived from diverse sources may contain different components. Collagens from finfish and shellfish are classified as major and minor types I and V, respectively [70]. Types I and V are ideal for biomedical applications because of their superior biocompatibility and low immunogenicity [71]. Fibril-forming collagens are categorized into Types I, II, III and V because a considerable percentage of homologous sequences are not species-dependent [72–74]. Collagen type I trimers consist of two α1 chains and one α2 chain, resulting in triple

helices that form banded fibrils [75–77]. This collagen type constitutes the primary protein component of the bone extracellular matrix, accounting for up to 90% of the organic matrix [78, 79]. These are necessary for bone tissue engineering [80]. It is an essential component of skin tissue and has been utilized for several purposes, including anti-inflammatory properties, skin enhancement, connective tissue strengthening and bone health [81, 82]. In most species, genes encoding for collagen type I trimers are *COL1A1* and *COL1A2*. The data reported in this study may facilitate further investigation of skin and skeletal disorders associated with abnormal type I collagen synthesis. However, in zebrafish, three type I collagen genes have been identified: *COL1A1a*, *COL1A1b*, and *COL1A2*, which encode α1(I), α2(I) and α3(I) chains, respectively [76].

The collagen type V trimer is a minor, low-abundance, yet crucial structural protein present in the extracellular matrix of diverse tissues that governs the morphology and tensile strength of collagen fibrils [83]. It is composed of three α1(V) chains and one α2(V) chain, referred to as the α1(V)2α2(V) heterotrimer. These chains form fibrils that regulate their diameter [70]. Imamura et al. [84] reported that *COL5A1*, *COL5A2*, and *COL5A3* encode collagen type V trimers. This heterotrimer is essential for dermal fibrillogenesis and aids in the formation of a functional skin matrix [85]. The main chain of type V collagen is *COL5A1*, which is expressed in the connective tissue surrounding ray fibroblasts and is partially connected to skeletal precursor cells [86]. It is found in several tissues, including skin, bones, muscles, and blood vessels. Nucleation of type I collagen fibrils relies on type V collagen fibrillar type I collagen assembly in the presence of fibronectin and integrins [87]. Collagen type V trimers promote cell migration and proliferation and participate in several critical biological processes [88–90]. In a joint effort, primary fibril-forming collagens (types I, II and III) together with a minor presence of collagen types V and XI form connective tissues, with types V and XI acting to regulate fibril-forming collagens [91].

Collagen types II, VI, VII, X, XI, XII, XVI, XVIII, and DCN are important proteins that confer structural support to many tissues, including skin, bone, cartilage, and tendons [92]. Collagen II is the key component of cartilage, required for maintaining its tensile strength and elasticity [11, 93]. It is also found in the eyes and other connective tissues [94]. Collagen VI connects the extracellular matrix to cells in several tissues such as muscles, cartilage, and skin [95]. Collagen VII is located in the basement membrane of epithelial cells and binds to the underlying tissues [77]. Collagen XI is present in the cartilage, bone, and cornea of the eye, and contributes to their structural stability. [96]. Furthermore, collagen coexists with type II collagen in cartilage tissues [97]. Collagen X plays a significant role in bone growth and mineralization as well as cartilage formation, particularly in jawed vertebrates [98, 99]. Tendons, ligaments, and other connective tissues contain collagen XII, which facilitates the anchoring of cells to the extracellular matrix [100]. Collagen XVI involves the formation of fibrils inside connective tissues and may also be found in the skin and other organs [101]. Collagen XVIII plays a role in angiogenesis and is present in the cornea of the eye [102]. DCN, or decorin, is a proteoglycan that modulates collagen fiber assembly and functionality across several tissues, such as skin, tendons, and cartilage [103].

While collagen genes in osteoblasts have been extensively studied in humans, mice, and chicks, little is known about these genes in fish [69]. Consequently, our findings concerning collagen-encoded genes expressed in snakehead caudal fins provide a novel avenue for in-depth research on cartilage- or bone-forming cells, with insights into the extracellular matrix composition in finned fish [86]. Moreover, our phylogenetic analysis identified similar marine resources for exploring the potential of collagen-base biomaterials, as highlighted by Senadheera et al. [104] and Pozzolini et al. [105]. The global marine collagen market was evaluated at USD 685 million and USD 633 million in 2020 and 2022, respectively, with projections

indicating growth to USD 1123 million by 2032 [106]. Collagen is highly sought after in industries such as cosmetics, food and nutraceuticals, wound healing, food packaging, pharmaceuticals and biomedicine, with the skin and other byproducts of cattle, pigs and birds serving as key sources [107]. Conversely, fish and other marine resources are gaining popularity [108], due to the reduced risk of disease transmission to humans from mammals and birds. Therefore, they may serve as alternatives to bovine collagen [41, 109]. In tissue engineering and regenerative medicine, type I collagen is widely utilized as the preferred biomaterial for manufacturing cell-instructive scaffolds [110, 111], which are artificial matrices that act as replacements for temporary extracellular matrices. These scaffolds aid in the delivery of specific signals to cells, hence stimulating cellular processes, such as tissue regeneration and matrix remodeling [97]. Improvement and sustained collagen extraction from marine and freshwater fish necessitates an understanding of the molecular basis of collagen synthesis and the identification of genetic resources [112]. Indeed, our present study on *C. argus* explains collagen regulation and may offer valuable molecular and genomic insights into a substantial volume of fish collagen.

Numerous studies have revealed the beneficial and functional properties of fish-derived collagen [9, 25, 42, 113]. Wound-healing properties have been observed in the common *C. argus* (group W) [114, 115]. In contrast, albino fish with a golden caudal fin (group J), utilized in the current investigation, showed wound healing characteristics [25]; yet, a greater number of genes are involved in collagen pathways compared to group W. Our findings on collagen-related genes may enhance the wound-healing capacity of collagen from this albino fish. Although we discovered that collagen-related genes are downregulated in golden albino *C. argus*, they may enrich collagen in areas exhibiting diverse gene expression patterns, necessitating further comprehensive studies. Sveen et al. similarly noted distinct collagen-related gene expression in the caudal fins of Atlantic salmon (*Salmo salar* L.) [116].

Several researchers have used marine-derived biomaterials [117–119]. Our current research has broadened the scope of further investigations to determine the physicochemical properties of collagen in *C. argus*, including its composition, extraction methods, processing factors, molecular and evolutionary foundations, as well as novel physical, chemical, and enzymatic changes in the collagen structure of this species, to develop suitable biomaterials for tissue engineering and other biomedical applications in near future.

## 5 Conclusion

This study examined two phenotypes of the heritable albino *C. argus* (gray-finned and golden-finned) and compared their DEGs to those of normal-morph individuals. The transcriptome sequencing of *C. argus* yielded 174,779 assembled sequences and 137,130 unigenes. In the golden-finned albino group, DEGs were mostly enriched in gene ontology related to cytoplasm and collagen, a finding that has not been previously documented in fish species. The heatmap indicates that the DEGs were significantly segregated, with black-white and gray-finned albinos clustering together, whereas golden-finned albinos were distinctly grouped apart. A total of 379 common DEGs were identified across all three groups. Furthermore, GO enrichment analysis revealed that both gray-finned and golden-finned albino individuals showed a greater number of downregulated DEGs compared to black-white individuals. In addition, DEGs were predominantly enriched in collagen-related pathways. The collagen components identified included collagen I of *COL1A1* and *COL1A2*, collagen II of *COL2A1*, collagen V of *COL5A1* and *COL5A2*, collagen VI of *COL6A1* and *COL6A3*, collagen IX of *COL9A3*, collagen X of *COL10A1*, collagen XI of *COL11A2*, collagen XII of *COL12A1*, collagen XVI of *COL16A1*, collagen XVIII of *COL18A1* and decorin (DCN), all of which play a role in

modulating the collagen matrix. Notably, the expression of these genes was reduced in golden albino *C. argus*, demonstrating a correlation between albinism and gene expression. These findings open avenues for further research on the regulation of collagen-related genes in fish. Fish collagen, derived from various sources, contains various components, with collagen types I and V being particularly desirable for biomedical applications. Our findings shed light on the genetic and molecular mechanisms underlying collagen metabolism in fish, which may lead to the development of novel biotechnological applications and provide an intuitive understanding of vertebrate regenerative capabilities.

## Supporting information

**S1 Table. Search results (Hit-tables) of homologous genes for each collagen-related gene of *C. argus* using blastn tool.**
(XLSX)

**S1 Fig. Percentage of completeness of the core set of genes in BUSCO for the assemblies obtained in this study.** Light blue: complete (C) and single copy genes (S); dark blue: complete (C) and duplicated genes (D); yellow: fragmented genes (F); red: missing genes (M).
(TIF)

**S2 Fig. Stackbar graphical representation on diversity of genes present in fish families homologous to collagen-related genes of *C. argus*.**
(TIFF)

## Acknowledgments

We acknowledge the support provided by the Wekemo Tech Group Co., Ltd. Shenzhen, China for sequencing and data analysis services. We are also thankful to Farhana Tasnim and Krishal Pradhan for their support in graph preparation, editing and illustration of the figures.

## Author Contributions

**Conceptualization:** Shixi Chen, Fardous Mohammad Safiul Azam, Yuanchao Zou.

**Data curation:** Shixi Chen, Li Ao, Rui Li.

**Formal analysis:** Shixi Chen, Na Li, Jianlan Wang.

**Funding acquisition:** Shixi Chen.

**Investigation:** Li Ao.

**Methodology:** Shixi Chen, Fardous Mohammad Safiul Azam.

**Resources:** Ning Li, Fardous Mohammad Safiul Azam, Yuanchao Zou, Rui Li.

**Supervision:** Yuanchao Zou.

**Validation:** Li Ao, Na Li, Jianlan Wang.

**Visualization:** Fardous Mohammad Safiul Azam, Na Li, Jianlan Wang.

**Writing – original draft:** Shixi Chen, Fardous Mohammad Safiul Azam.

**Writing – review & editing:** Shixi Chen, Ning Li, Fardous Mohammad Safiul Azam, Zakaria Hossain Prodhan.

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
