## [Decision Letter · Decision Letter 0]

21 Feb 2024

PONE-D-23-30448Comparative transcriptome analysis of albino northern snakehead (Channa argus) reveals its distinct collagen-enriched DEGs in caudal fin cells: a resource for biomedical usePLOS ONE

Dear Dr. Safiul Azam,

Thank you for submitting your manuscript to PLOS ONE. After careful consideration, we feel that it has merit but does not fully meet PLOS ONE’s publication criteria as it currently stands. Therefore, we invite you to submit a revised version of the manuscript that addresses the points raised during the review process.

**ACADEMIC EDITOR: **In my opinion and based on the reviewers` comments I suggest the deep revision of the paper. I would like to give the opportunity to the Authors to make a correction of their manuscript. The main concern is the methodology, while the second  is that the conclusions are not supported by the results. Thus all these aspects should be improved. Please make a correction according to the attached comments of the Reviewers.

We look forward to receiving your revised manuscript.

Kind regards,

Karolina Goździewska-Harłajczuk

Academic Editor

PLOS ONE

Journal Requirements:

https://linkinghub.elsevier.com/retrieve/pii/S2405844023021448

https://www.tandfonline.com/doi/abs/10.1080/10408398.2020.1751585

In your revision ensure you cite all your sources (including your own works), and quote or rephrase any duplicated text outside the methods section. Further consideration is dependent on these concerns being addressed.

4. In the online submission form, you indicated that the data supporting these findings will be available in online repositories. The names of the repository/repositories and accession numbers are as follows: Bioproject number PRJNA913664 and BioSample number SAMN32303246, which will be released on October 5, 2023. Before the above date, the raw data supporting the conclusions of this article will be made available by the authors upon reasonable request without undue reservation.

5. We notice that your supplementary figures are included in the manuscript file. Please remove them and upload them with the file type 'Supporting Information'. Please ensure that each Supporting Information file has a legend listed in the manuscript after the references list.

Additional Editor Comments:

In my opinion and based on the reviewers` comments I suggest the deep revision of the paper. I would like to give the opportunity to the Authors to make a correction of their manuscript. The main concern is the methodology, while the second is that the conclusions are not supported by the results. Thus all these aspects should be improved. Please make a correction according to the attached comments of the Reviewers.

Reviewers' comments:

Reviewer's Responses to Questions

**Comments to the Author**

1. Is the manuscript technically sound, and do the data support the conclusions?

Reviewer #1: No

Reviewer #2: Yes

2. Has the statistical analysis been performed appropriately and rigorously? 

Reviewer #1: N/A

Reviewer #2: Yes

3. Have the authors made all data underlying the findings in their manuscript fully available?

Reviewer #1: Yes

Reviewer #2: Yes

4. Is the manuscript presented in an intelligible fashion and written in standard English?

Reviewer #1: No

Reviewer #2: Yes

5. Review Comments to the Author

Reviewer #1: The authors performed transcriptome sequencing of caudal fins to compare normal and albino northern snakehead. Although numerous data were generated, the overall writing of this manuscript is poor, and the conclusions or manuscript title cannot be supported by these data. The reviewer would recommend rejection of the present version of this manuscript.

Extra editing is required since the overall writing is poor.

Line 123: Provide a reference for “the ARRIVE guidelines”.

Fig 1: Pick out one fish image as a representative for each group.

Lines 127-136: Provide more details (such as city and country names) for the companies when necessary.

Line 150: Provide the full name for TPM and a reference for “RSEM software”.

Lines 159-162: Move this figure to the supplementary section.

Line 164: Rewrite this subtitle.

Lines 173-174: Change “GB” to “Gb”. What is “CDS predictions”?

Table 1: Add more data for those missing details.

Lines 182-185: Rewrite these sentences.

Lines 228-230: Why? It doesn’t make sense.

Lines 265-270: The authors may provide more details about these collagen genes, such as transcription difference (ratios) and protein sequences. Is these any premature termination or mismatch of certain important DEG? This may explain the albinism color.

Discussion: These documents can not support the manuscript even the title, since DEG differences or biomedical uses are missing.

Supplementary Fig 1: Why are the percentages of complete BUSCOs too low?

Reviewer #2: Manuscript can be accepted after minor changes

I am of the opinion that if the keywords include words other than the words in the title, it may increase the citation and recognition of the article.

Complete line 147 in manuscript file.

6. PLOS authors have the option to publish the peer review history of their article (what does this mean?). If published, this will include your full peer review and any attached files.

Reviewer #1: No

Reviewer #2: No

---

## [Author Response · Author response to Decision Letter 0]

15 Aug 2024

The response to Editor and the reviewer's comment, both as been uploaded in the system.

---

## [Decision Letter · Decision Letter 1]

15 Sep 2024

PONE-D-23-30448R1Comparative transcriptome analysis of albino Northern Snakehead (Channa argus) reveals its downregulated collagen-enriched DEGs in caudal fin cellsPLOS ONE

Dear Dr. Safiul Azam,

Thank you for submitting your manuscript to PLOS ONE. After careful consideration, we feel that it has merit but does not fully meet PLOS ONE’s publication criteria as it currently stands. Therefore, we invite you to submit a revised version of the manuscript that addresses the points raised during the review process.

**The manuscript was corrected according to the reviewers` suggestions, however the paper still needs some revision.**

We look forward to receiving your revised manuscript.

Kind regards,

Karolina Goździewska-Harłajczuk

Academic Editor

PLOS ONE

**Additional Editor Comments:**

The manuscript was corrected according to the reviewers` suggestions, however the paper still needs some revision.

Reviewers' comments:

Reviewer's Responses to Questions

**Comments to the Author**

1. If the authors have adequately addressed your comments raised in a previous round of review and you feel that this manuscript is now acceptable for publication, you may indicate that here to bypass the “Comments to the Author” section, enter your conflict of interest statement in the “Confidential to Editor” section, and submit your "Accept" recommendation.

Reviewer #1: (No Response)

Reviewer #2: All comments have been addressed

2. Is the manuscript technically sound, and do the data support the conclusions?

Reviewer #1: Partly

Reviewer #2: Yes

3. Has the statistical analysis been performed appropriately and rigorously? 

Reviewer #1: Yes

Reviewer #2: Yes

4. Have the authors made all data underlying the findings in their manuscript fully available?

Reviewer #1: No

Reviewer #2: (No Response)

5. Is the manuscript presented in an intelligible fashion and written in standard English?

Reviewer #1: No

Reviewer #2: Yes

6. Review Comments to the Author

**Reviewer #1: **The authors performed a comparative transcriptome analysis to revela down-regulated college –enriched DEGs in the caudal fins of albino Northern snakehead. It seems that collagen gene expression is limited. In general, the overall writing of this manuscript can be improved with extra editing, although numerous data were generated to support the main conclusions.

1. Why did the authors select caudal fins? It would be much better to collect muscle samples for transcriptome sequencing and comparative analysis. The authors should provide some descriptions in the Introduction or Discussion.

2. Rewrite the subtitles in the Results section, since they look more like those for Materials and Methods.

3. Although the authors performed transcriptome sequencing in triplications, they are recommended to validate at least collage expression by additional qRT-PCR.

4. Make sure that related transcriptome data are released.

**Reviewer #2: **The manuscript is accepted in its current form. The abstract is clear and concise, effectively summarizing the key points. The introduction provides a thorough overview of the subject matter, and the literature review is both comprehensive and well-integrated into the discussion. Each section of the paper is well-explained, leaving no unanswered questions. Overall, the manuscript is well-structured and demonstrates a strong grasp of the topic.

7. PLOS authors have the option to publish the peer review history of their article (what does this mean?). If published, this will include your full peer review and any attached files.

Reviewer #1: No

Reviewer #2: No

---

## [Author Response · Author response to Decision Letter 1]

30 Nov 2024

Response to the Reviewers:

Thanks to the reviewers for the comments and suggested revisions to this manuscript. The author's responses are shared below in a table, and the manuscript is revised according to the track changes. 

Moreover, we have made an extensive English language improvement in the manuscript as per advice from the Editor.

Apart from that, we did a minor revision to the title of our manuscript. Moreover, we have revised Figure 9 (made scientific names in italics) and then uploaded the new image into the system.

Moreover, the lines mentioned (as per with track change mood version) in the table below are according to the revised manuscript. 

Reviewer 1 

Reviewer’s Comments 1 : Why did the authors select caudal fins? It would be much better to collect muscle samples for transcriptome sequencing and comparative analysis. The authors should provide some descriptions in the Introduction or Discussion 

Response from Author: Thanks for your comment. Our study focuses on the distinctive characteristics of the color morph of C. argus, specifically targeting the unique color differences in their fins. Although previous research has explored various aspects of this species—such as whole genome analysis, genome characterization, molecular diversity, disease and pathogenesis, skin color and albinism, sex chromosomes, feeding behavior, and surimi products—none have investigated the specific color variation in the most manifest part of caudal fins. To address this gap, we collected caudal fin tissues, intentionally excluding scales, skin, or muscle. The golden-finned C. argus stands out from other color morphs, and recent studies have identified fin tissues as a promising source of collagen. We have elucidated this in the Introduction section. 

Please see lines: 575 to 836.

Reviewer’s Comments 2: Rewrite the subtitles in the Results section, since they look more like those for Materials and Methods Response from Author: Thanks for your comment. We have revised them all.

Reviewer’s Comments 3: Although the authors performed transcriptome sequencing in triplications, they are recommended to validate at least collagen expression by additional qRT-PCR. 

Response from Author: Thanks for your comment. We appreciate the reviewer's suggestion to validate collagen expression with additional qRT-PCR. However, due to resource constraints—including limited access to fish samples and funding—we are unable to conduct further experiments. Our study utilized robust transcriptome sequencing in triplicate, ensuring reliable data for collagen expression analysis. Additionally, we have established thorough phylogenetic evidence to support our findings, confirming the evolutionary relevance and consistency of the observed patterns. We are confident that the combination of high-quality transcriptomic data and phylogenetic analysis provides a solid foundation for our conclusions. Moreover, collagen-enriched caudal fins were also been observed in Atlantic salmon by Saveen et al. (2023). Please see lines: 4142 to 4144. 

Reviewer’s Comments 4: Make sure that related transcriptome data are released. 

Response from Author: Thanks for your comment. We have deposited and released the data in NCBI under Bioproject number PRJNA913664 and BioSample number SAMN32303246 with accession number of SRX18894708-SRX18894716. Please see lines: 4689 to 4692.

Reviewer 2: No comments observed -

Thank you.

Sincerely yours,

Shixi Chen, Ph.D. and Fardous Mohammad Safiul Azam, Ph.D. 

College of Life Sciences, Neijiang Normal University, Neijiang, 641100, China.

---

## [Decision Letter · Decision Letter 2]

5 Dec 2024

Comparative transcriptome analysis of albino Northern Snakehead (Channa argus) reveals its various collagen-related DEGs in caudal fin cells

PONE-D-23-30448R2

Dear Dr. Azam,

We’re pleased to inform you that your manuscript has been judged scientifically suitable for publication and will be formally accepted for publication once it meets all outstanding technical requirements.

Kind regards,

Karolina Goździewska-Harłajczuk

Academic Editor

PLOS ONE

Additional Editor Comments (optional):

All comments have been addressed, thus the manuscript can be accept in its current form.

Reviewers' comments:

Reviewer's Responses to Questions

**Comments to the Author**

1. If the authors have adequately addressed your comments raised in a previous round of review and you feel that this manuscript is now acceptable for publication, you may indicate that here to bypass the “Comments to the Author” section, enter your conflict of interest statement in the “Confidential to Editor” section, and submit your "Accept" recommendation.

Reviewer #1: All comments have been addressed

2. Is the manuscript technically sound, and do the data support the conclusions?

Reviewer #1: Yes

3. Has the statistical analysis been performed appropriately and rigorously? 

Reviewer #1: N/A

4. Have the authors made all data underlying the findings in their manuscript fully available?

Reviewer #1: Yes

5. Is the manuscript presented in an intelligible fashion and written in standard English?

Reviewer #1: Yes

6. Review Comments to the Author

Reviewer #1: The authors made reasonable responses and corrections in the revised manuscript. The present version of this manuscript is acceptable.

7. PLOS authors have the option to publish the peer review history of their article (what does this mean?). If published, this will include your full peer review and any attached files.

Reviewer #1: No

---

## [Editor Report · Acceptance letter]

15 Dec 2024

PONE-D-23-30448R2 

PLOS ONE

Dear Dr. Safiul Azam, 

I'm pleased to inform you that your manuscript has been deemed suitable for publication in PLOS ONE. Congratulations! Your manuscript is now being handed over to our production team.

Kind regards, 

on behalf of

Dr. Karolina Goździewska-Harłajczuk 

Academic Editor

PLOS ONE